# Therapeutic Targeting of Ovarian Cancer Stem Cells Using Estrogen Receptor Beta Agonist

**DOI:** 10.3390/ijms23137159

**Published:** 2022-06-28

**Authors:** Yi He, Salvador Alejo, Prabhakar Pitta Venkata, Jessica D. Johnson, Ilanna Loeffel, Uday P. Pratap, Yi Zou, Zhao Lai, Rajeshwar R. Tekmal, Edward R. Kost, Gangadhara R. Sareddy

**Affiliations:** 1Department of Obstetrics and Gynecology, University of Texas Health San Antonio, San Antonio, TX 78229, USA; 2204130809@csu.edu.cn (Y.H.); alejo@livemail.uthscsa.edu (S.A.); prabhakarp@uthscsa.edu (P.P.V.); johnsonj20@livemail.uthscsa.edu (J.D.J.); idinesman@gmail.com (I.L.); pratap@uthscsa.edu (U.P.P.); tekmal@uthscsa.edu (R.R.T.); kost@uthscsa.edu (E.R.K.); 2Department of Neurosurgery, Xiangya Hospital, Central South University, Changsha 410008, China; 3Greehey Children’s Cancer Research Institute, University of Texas Health San Antonio, San Antonio, TX 78229, USA; zou@uthscsa.edu (Y.Z.); laiz@uthscsa.edu (Z.L.); 4Mays Cancer Center, University of Texas Health San Antonio, San Antonio, TX 78229, USA

**Keywords:** estrogen receptor beta, ovarian cancer, cancer stem cells, LY500307, apoptosis

## Abstract

Ovarian cancer (OCa) is the deadliest gynecologic cancer. Emerging studies suggest ovarian cancer stem cells (OCSCs) contribute to chemotherapy resistance and tumor relapse. Recent studies demonstrated estrogen receptor beta (ERβ) exerts tumor suppressor functions in OCa. However, the status of ERβ expression in OCSCs and the therapeutic utility of the ERβ agonist LY500307 for targeting OCSCs remain unknown. OCSCs were enriched from ES2, OV90, SKOV3, OVSAHO, and A2780 cells using ALDEFLUOR kit. RT-qPCR results showed ERβ, particularly ERβ isoform 1, is highly expressed in OCSCs and that ERβ agonist LY500307 significantly reduced the viability of OCSCs. Treatment of OCSCs with LY500307 significantly reduced sphere formation, self-renewal, and invasion, while also promoting apoptosis and G2/M cell cycle arrest. Mechanistic studies using RNA-seq analysis demonstrated that LY500307 treatment resulted in modulation of pathways related to cell cycle and apoptosis. Western blot and RT-qPCR assays demonstrated the upregulation of apoptosis and cell cycle arrest genes such as FDXR, p21/CDKN1A, cleaved PARP, and caspase 3, and the downregulation of stemness markers SOX2, Oct4, and Nanog. Importantly, treatment of LY500307 significantly attenuated the tumor-initiating capacity of OCSCs in orthotopic OCa murine xenograft models. Our results demonstrate that ERβ agonist LY500307 is highly efficacious in reducing the stemness and promoting apoptosis of OCSCs and shows significant promise as a novel therapeutic agent in treating OCa.

## 1. Introduction

Ovarian cancer (OCa) is the fifth leading cause of cancer-related deaths in the United States and is the deadliest of all gynecologic cancers [1,2]. With an overall 5-year survival rate of approximately 50%, patients face a poor prognosis [3]. Recent figures from the American Cancer Society estimate an incidence of 21,410 new OCa cases in 2021, alongside an estimated 13,770 deaths from OCa. Accounting for these grim statistics, OCa lacks early-stage markers and is typically diagnosed in advanced stages [4]. OCa is a heterogenous disease that can be subdivided into different histological subtypes. Epithelial ovarian cancers (EOC) account for ~90% of OCa which includes serous, endometrioid, clear-cell, and mucinous carcinomas [4]. Standard treatment for newly diagnosed OCa consists of cytoreductive surgery and platinum-based chemotherapy. Although OCa patients initially respond to these regimens, nearly 90% will develop recurrence and, inevitably, succumb to chemotherapy-resistant disease [5]. Several lines of evidence suggest cancer stem cells (CSCs), a subpopulation of tumor cells with self-renewal capacity, are implicated in the tumor initiation and chemoresistance of OCa [6,7,8,9]. The enrichment of CSCs after chemotherapy has been demonstrated in preclinical OCa models and OCa patient samples, supporting the premise that ovarian CSCs contribute to disease recurrence [6,10,11]. There is a critical need to develop safe and effective therapeutic strategies that eradicate ovarian cancer stem cells (OCSCs) to overcome chemoresistance and prevent tumor recurrence in OCa [12,13].

Estrogens are the main female sex steroidal hormones that play a critical role in the proliferation and differentiation of the ovary [14]. The biological effects of estrogens are primarily mediated through estrogen receptor alpha (ERα/ESR1) and estrogen receptor beta (ERβ/ESR2) [15,16]. Unlike ERα, ERβ has contrasting functions as a tissue-specific tumor suppressor that promotes anti-proliferative actions [16]. ERβ exerts its functions as either a homodimer (ERβ/ERβ) or heterodimer (ERα/ERβ) depending on the status of the cellular expression of ERs [17,18]. Heterodimerization of ERβ with ERα results in blunting of ERα mediated proliferative actions [17,19]. ERβ functions through three different cellular mechanisms to mediate its biological responses that include classical, non-classical, and extra-nuclear signaling [20]. Classical signaling involves direct binding of ERβ to target gene promoters that consists of estrogen response elements (EREs). ERβ mediated effects can be non-classical via its interactions with other transcription factors such as AP1, SP1, and KLF5, and ERβ also promote extra-nuclear signaling by interacting with cytosolic kinases [20,21].The ERβ expression is downregulated in many tumors including those of the breast, ovary, prostate, colon, and brain [22,23,24,25,26]. The decrease in ERβ expression increases the risk for metastasis in OCa [27] and correlates with a lack of clinical response to chemotherapy and shorter overall survival in OCa [28]. Further, reintroduction of ERβ in OCa cells reduces OCa cell proliferation [28]. 

Although estrogen (E2) can stimulate ERβ, the potential of E2 as therapy has limited use due to its suspected role in breast cancer etiology via stimulation of ERα. Further, the clinical utility of targeting ERβ is limited due to a lack of mechanistic insights and availability of agents that specifically target ERβ. Even though ERα and ERβ are structurally similar, their ligand-binding domains differ enough to be selective for different ligands [21,29]. Several selective synthetic or natural ERβ agonists have been identified and are being investigated for therapeutic use. LY500307 (SERBA-1, erteberel) is a potent (EC_50_ = 0.66 nM) and selective synthetic ERβ agonist developed by Eli Lilly and Company [30]. LY500307 has a 14-fold higher affinity for ERβ than ERα, and exhibits 32-fold more potency in functional assays. LY500307 was well tolerated in benign prostatic hypertrophy (BPH) patients with no side effects [31], and has, also, been tested in phase 2 clinical trials for improving negative symptoms and cognitive impairment associated with Schizophrenia (http://clinicaltrials.gov/show/NCT01874756, accessed on 25 December 2021). Further, LY500307 was shown to exhibit antitumor functions in glioblastoma models [32,33] and activation of ERβ with LY500307 augments innate immunity to suppress lung metastasis of triple negative breast cancer and melanoma cells [34]. However, the role of ERβ in OCSCs and the effect of ERβ agonist on OCSCs functions remain largely unknown. 

In this study, we investigated the significance of ERβ in OCSCs, and tested the efficacy of ERβ agonist LY500307 in reducing OCa stemness. Our results demonstrated that ERβ, in particular, ERβ isoform 1, is highly expressed in OCSCs and that ERβ agonist LY500307 potently reduced the cell viability, sphere formation, and self-renewal of OCSCs. Transcriptomic analysis showed LY500307 increased expression of tumor suppressor genes such as CDKN1A and FDXR. Western blot analysis showed that LY500307 treatment reduced the expression of stemness markers and increased the expression of apoptotic markers. Further, LY500307 treatment reduced the tumor initiation capacity of OCSCs in orthotopic OCa xenograft models.

## 2. Results

### 2.1. Estrogen Receptor β Agonist Reduces the Viability of OCSCs

Several studies demonstrated that ERβ is highly expressed in a normal ovary, and its expression is significantly reduced during OCa progression [27,35,36,37]. However, it is not known whether ERβ is expressed in OCSCs and whether ERβ agonists have therapeutic utility in eradication of OCSCs. To address this gap, OCSCs were enriched using established stem cell marker ALDH [38,39,40,41] from ES2, OV90, and OVSAHO OCa cell lines using ALDEFLUOR assay. We determined the expression status of ERβ isoforms in OCSCs and non-OCSCs. RT-qPCR results showed that ERβ1, which is generally considered as a tumor suppressor, is expressed in both non-OCSCs and OCSCs, however, its expression is significantly higher in OCSCs compared to non-OCSCs (Figure 1A–C). Further, we also determined the expression of ERβ isoforms in ES2, SKOV3, and A2780 stem-like (spheroids) and non-stem like (adherent monolayer) cells. Consistent with our results in ALDH+ve OCSCs, OCa spheroids highly express ERβ1 compared to adherent OCa cell lines (Figure 1D–F). However, no significant trend was observed in the expression of ERβ2 and ERβ5 isoforms in OCSCs. These results suggest that ERβ1 is highly expressed in OCSCs enriched from multiple OCa cell lines. Next, we examined ERβ protein expression using immunocytochemistry (ICC). Since SKOV3 and A2780 cells are the widely used OCa cell lines, we used these cells in most experiments. As shown in Figure 1G,H, both ALDH-ve and ALDH+ve cells enriched from SKOV3 and A2780 cells express ERβ, albeit a higher level of expression is detected in ALDH+ve cells compared to ALDH-ve cells. Since ERβ1, a full-length ligand binding ERβ isoform, is highly expressed in OCSCs, we asked whether stimulation of ERβ with agonist causes a reduction in viability of OCSCs. Our results show that ERβ agonist LY500307 treatment significantly reduced the viability of ALDH+ve OCSCs compared to ALDH-ve non-OCSCs (Figure 1I,J). These results suggested that ERβ is highly expressed in OCSCs and ERβ agonist significantly reduced the viability of OCSCs.

### 2.2. ERβ Agonist LY500307 Reduces Sphere Formation, Self-Renewal, and Invasion and Induces Apoptosis and G2/M Cell Cycle Arrest in OCSCs

To determine whether ERβ agonist LY500307 affects the sphere formation ability of OCSCs, SKOV3-, and A2780- ALDH+ve cells were treated with LY500307 for 7 days and the number of newly formed spheres were counted. As shown in Figure 2A,B, LY500307 treatment significantly reduced the sphere formation of OCSCs compared to vehicle treatment. Further analysis indicated LY500307 treatment reduced the growth of spheres which is evident from the reduced sphere size compared to vehicle. To determine the effect of LY500307 on the self-renewal ability of OCSCs, we performed limiting dilution assays. SKOV3- and A2780- ALDH+ve cells were seeded in decreasing numbers and treated with LY500307. After 10 days, spheres formed per each plating density was determined, and the stem cell frequency was analyzed using ELDA analysis software. Treatment with LY500307 significantly reduced the self-renewal ability of OCSCs compared to vehicle (Figure 2C,D). To further examine the effect of LY500307 on apoptosis, SKOV3- and A2780- ALDH+ve cells were treated with vehicle or LY500307 and apoptosis was measured by Annexin V/PI assay. As shown in Figure 2E–H, LY500307 treatment significantly induced apoptosis of OCSCs when compared to vehicle. To determine the effect of LY500307 on cell cycle progression of OCSCs, we performed flow cytometry analysis of PI-stained cells. Results showed LY500307 treatment significantly increased the proportion of OCSCs in G2/M cell cycle phase (Figure 2I–L). Further, to test whether LY500307 treatment affects invasion of OCSCs, matrigel invasion assays were performed. As shown in Figure 2M, LY500307 treatment significantly decreased the invasion ability of OCSCs when compared to vehicle (Figure 2M). Altogether, these results suggest that ERβ agonist treatment reduced the self-renewal capacity of OCSCs while promoting apoptosis, reducing invasion, and arresting cells in G2/M phase.

### 2.3. Analysis of Transcriptional Changes Altered by LY500307 Treatment in OCSCs

To examine the global transcriptional changes in OCSCs following LY500307 treatment, we performed RNA-sequencing on OCSCs treated with either vehicle or LY500307. Compared with vehicle, 596 genes were identified as being differentially expressed in LY500307-treated OCSCs, including 382 upregulated genes and 214 downregulated genes (Appendix A). A volcano plot of differentially expressed genes (DEGs) is shown in Figure 3A and a representative heat map of top DEGs is shown in Figure 3B. To further examine the biological significance of LY500307 regulated genes while eliminating the bias from artificial selection criteria for DEGs, all the expressed genes, instead of only DEGs, were utilized for functional enrichment with GSEA analysis. We found that LY500307-regulated genes showed positive enrichment in the p53 pathway, oxidative phosphorylation, apoptosis, and estrogen late response genes, and negative correlation with E2F targets, G2/M checkpoint, and mitotic spindle. Top LY500307-regulated hallmark gene sets were shown using a bubble plot (Figure 3C). The enrichment plots of apoptosis, p53 pathway, cell cycle, and G2/M checkpoint are shown in Figure 3D–G. We found that several genes involved in cell cycle arrest and apoptosis such as CDKN1A, FDXR, TP53I3, ANXNA1, PLTP, and SESN2, were upregulated, and many genes involved in CSCs and oncogenic functions, such as KCND1, PDK1, DOK3, AGER, CFH, DNHD1, MYBL1, HK2, and STC2, were downregulated. We further validated these findings in SKOV3- and A2780- OCSCs using RT-qPCR (Figure 3H,I).

### 2.4. ERβ Agonist Induces the Expression of FDXR and CDKN1A in OCSCs

For an unbiased exploration of ERβ direct targets, we further performed in silico analyses using multiple online databases and datasets. Two ERβ specific motifs were acquired from JASPAR online platform and mapped with the FIMO and T-Gene modules from the MEME Suite in sequence to identify potential binding target genes (Figure 4A). A total of 7017 and 8481 genes were identified as binding targets for ERβ motif 1 and motif 2, respectively. Thereafter, through the intersection of those potential target genes with LY500307-regulated DEGs, we identified a total of 127 candidate genes (Appendix A), including the top gene FDXR, as potential ERβ target genes in OCSCs (Figure 4B). Our RNA-sequencing data showed that LY500307 treatment increased the expression of FDXR and CDKN1A in OCSCs. To further verify whether FDXR and CDKN1A are direct targets of ERβ, we utilized published ChIP-seq profiles in CistromeDB to confirm the enrichment of ERβ binding at FDXR and CDKN1A promoters (Figure 4C,D). In addition, gene co-expression analysis using TCGA OV tumor dataset with GEPIA2 indicated that the expression level of ERβ positively correlated with FDXR and CDKN1A (Figure 4E,F). Next, we investigated whether LY500307 activates ERβ transactivation in OCSCs using ERE-luc reporter assays. As shown in Figure 4G, ERβ agonist LY500307 significantly induced the ERE-luc reporter activity in OCSCs. To determine whether ERβ agonist treatment increases CDKN1A (p21 protein) and FDXR expression, we treated OCSCs with LY500307 and analyzed their expression using Western blotting. Results showed LY500307 treatment significantly increased expression of p21 and FDXR in both SKOV3- and A2780-ALDH+ve OCSCs (Figure 4H,I). These results suggest that FDXR and p21/CDKN1A are direct targets of ERβ, and that ERβ agonist LY500307 promotes their expression. Since LY500307 treatment reduced the stemness and promoted the apoptosis of OCSCs, we sought to explore its effect on the expression of stemness and apoptotic markers. RT-qPCR and Western blot results showed that treatment of OCSCs with LY500307 resulted in a significant reduction of stemness markers SOX2, CD133, Oct4, and Nanog (Figure 4J–L). Further, Western blot results showed that LY500307 treatment substantially increased levels of cleaved PARP and cleaved caspase 3 in OCSCs (Figure 4M). Collectively, these results suggest that agonist stimulation of ERβ resulted in reduction of stemness and induction of apoptotic markers in OCSCs.

### 2.5. ERβ Agonist Treatment Suppresses Tumor Initiation Capacity of OCSCs in Orthotopic Models

Since in vitro results demonstrated that LY500307 treatment decreases the stemness of OCSCs, we further evaluated the effect of LY500307 treatment on OCSCs driven tumor initiation in vivo using an orthotopic intrabursal xenograft model. SKOV3-ALDH+ve OCSCs that were transduced with GFP-Luc, were treated with vehicle or LY500307, and injected viable OCSCs into ovarian bursa of SCID mice in various dilutions. Tumor development was monitored weekly for 30 days using Xenogen IVIS. As shown in Figure 5A, in vehicle treated group, 4/4 (100%) mice injected with 1 × 10^6^ cells and 3/4 mice (75%) injected with 1 × 10^5^ cells showed tumor establishment. In the LY500307 treatment group, only 3/4 (75%) of mice injected with 1 × 10^6^ cells developed tumors (Figure 5A). None of the mice developed tumors injected with 1 × 10^5^ cells. ELDA analysis of tumor initiation data is shown in Figure 5B. Representative bioluminescent images of mice injected with vehicle or LY500307 treated OCSCs are shown (Figure 5C,D). These results suggested that ERβ agonist LY500307 reduced the stemness of OCSCs leading to reduced tumor initiation.

## 3. Discussion

Although ERβ is an established tumor suppressor in OCa, its role in stemness of OCSCs has not been previously elucidated. In this study, we provided evidence that ERβ, particularly ERβ isoform 1, is highly expressed in OCSCs and that treatment with ERβ agonist LY500307 potently reduced the cell viability of OCSCs. In addition, LY500307 treatment attenuated sphere formation and self-renewal capacity of OCSCs, while also inducing apoptosis and cell cycle arrest. Mechanistic studies demonstrate that ERβ agonist treatment increased expression of tumor suppressor genes FDXR and CDKN1A/p21 in OCSCs. Further, we show LY500307 has the potential to attenuate OCSCs mediated tumor initiation in vivo. 

Emerging studies demonstrated the involvement of ERβ in stem cells. ERβ is expressed in human prostate stem/progenitor cells enriched from cancerous tissues [42] and activation of ERβ impairs prostatic regeneration by inducing apoptosis in stem/progenitor cells [43]. Further, agonist mediated activation of ERβ attenuates self-renewal of murine prostatic stem/progenitor cells [43]. ERβ overexpression or activation reduced stemness and induced the apoptosis of glioma stem cells [33]. Breast cancer stem cells (BSCs) express ERβ, and upregulation of ERβ in BSCs was associated with phenotypic stem cell markers [44], suggesting tissue specific effects of ERβ. A recent study showed that diosmetin, an O-methylated flavanone targets ERβ and ERβ expression conferred cell sensitivity, as patient-derived AML cells with high levels of ERβ were sensitive, whereas cells with low ERβ were less sensitive to diosmetin [45]. Our results corroborate these findings that OCSCS express ERβ and activation of ERβ reduced the viability of OCSCs.

ERβ exists in five different isoforms which exhibit distinct roles in different cancers [46]. The full-length ligand activated ERβ1 is widely regarded as a tumor suppressor; however, other isoforms, in particular, ERβ2 and ERβ5, are overexpressed in several cancers and associated with poor prognosis of breast, prostate, and ovarian cancer patients [47,48,49,50,51]. Interestingly, ERβ2 and ERβ5 isoforms induce stem cell characteristics and chemotherapy resistance in prostate cancer through activation of hypoxic signaling [52]. In our study, we observed that ERβ1, which is a full-length ligand binding isoform of ERβ, is highly expressed in OCSCs. Further, treatment with ERβ agonist LY500307 reduced the sphere formation and self-renewal, as well as promoted the apoptosis and G2/M cell cycle arrest of OCSCs. Importantly, treatment with ERβ agonist LY500307 attenuated the in vivo tumor initiating capacity of OCSCs in mouse xenograft models. Since ERβ isoforms expression is tissue- and cell-specific and exerts different functions, and the levels of other isoforms dictate the function of tumor suppressive ERβ1, it is likely that it may have distinct roles in the CSCs phenotype, as well as in a tissue specific manner. Further counteracting oncogenic functions of other ERβ isoforms and ratio of levels of ERβ1 with other ERβ isoforms may have implications in the CSCs phenotypes. In our study, we observed that non-OCSCs are also sensitive to ERβ agonist treatment, and they do express ERβ. However, we found that OCSCs express higher levels of ERβ1 than non-OCSCs, and this may correlate with the increased activity of ERβ agonist LY500307. However, we cannot rule out that the mechanism of action by which ERβ agonist acts may be different in OCSCs and non-OCSCs populations.

Previous studies showed that ERβ is highly expressed in a normal ovary, and its expression is significantly reduced during ovarian cancer progression, however, it is not clear when and how does OCSCs exhibit the expression of ERβ. We speculate that differential ERβ expression in bulk tumor cells vs. OCSCs could be due to altered expression of ERβ isoforms and cell/tissue specific function of ERβ. Further, some evidence implicated the lack of clinical response to chemotherapy due to decreased ERβ expression which is attributed, in part, due to increased expression of other isoforms such as ERβ2, ERβ5 that contribute to chemotherapy resistance. Future studies are clearly needed to understand how ERβ is regulated in bulk tumor cells and OCSCs. Interestingly, a high level of ERβ, and not ERα, expression in stage III HGSOC patients can predict the efficacy of platinum plus taxane chemotherapy. Compared to ERβ low expression group, the high ERβ expression group exhibited a 2 times greater median progression-free survival and a 2.2 times less recurrence risk [53], and the status of ERβ can be considered as one of the possible predictors for evaluating the effectiveness of OCa therapy [53]. Importantly, knockdown of ERβ confirmed resistance, whereas overexpression enhanced sensitivity to diosmetin [45]. These studies further support our findings that ERβ could be considered as a target for CSCs enriched disease such as chemotherapy resistant OCa. Since, ERβ expression is elevated in OCSCs, the treatment of OCSCs with ERβ agonist LY500307 alone or in combination with chemotherapy may be a potentially effective strategy for chemotherapy resistant disease. A limitation of this study is that we enriched cells for CSCs properties using an ALDH marker, and our conclusions are limited to ALDH+ve cells. Future studies are clearly needed to establish whether ERβ agonists can address chemotherapy resistance in OCa and can act on other CSCs subpopulations.

Several studies conducted transcriptomic studies to identify the ERβ target genes that contribute to inhibition of cancer cell growth and induction of apoptosis. Our GSEA analysis results showed that apoptosis pathway gene sets are positively enriched in the LY500307-treated group. In this study, using the ERβ agonist mediated transcriptome, we identified that FDXR and CKDN1A are among the top upregulated genes in OCSCs. It has been shown that for CDKN1A (p21) promoter region has several estrogen response element (ERE) half-sites and is directly regulated by ERβ [54,55]. Several studies have shown that ERβ inhibits the proliferation of cancer cells via upregulation of the expression of p21 [56,57,58,59]. Importantly, studies showed that p21 controls the expansion of stem cells [60,61] as well as cancer stem-like cells [62]. Consistent with these studies we also observed that ERβ agonist increased the expression of p21 in OCSCs. We also observed the downregulation of several genes involved in CSCs functions including PDK1, MYBL1, HK2, and STC2 supporting the ERβ agonist mediated reduction of sphere formation and self-renewal of OCSCs.

FDXR is a mitochondrial membrane-associated flavoprotein that initiates the mitochondrial electron transport chain by transferring electrons from NADPH to the mitochondrial cytochrome P450 system via the ferredoxins FDX1 or FDX2. This pathway also plays a critical role in biosynthesis of steroid hormones and drug metabolism pathways [63]. Mutations in *FDXR* causes sensorial neuropathies [64], optic atrophy [65], and neurodegeneration associated with inflammation [66]. FDXR is a p53 regulated gene, and is essential for p53 mediated apoptosis through generation of oxidative stress in the mitochondria in colorectal cancer cells [67]. Further, overexpression of FDXR increases the sensitivity of tumor cells to apoptosis [68]. Our results suggest that ERβ agonist LY500307 treatment increases the FDXR expression in OCSCs. Importantly, ERβ binding motifs are enriched in FDXR gene, and using published ChIP-seq profiles we confirmed that ERβ is recruited to the promoter of FDXR gene. Further, our study also found a positive correlation of ERβ and FDXR in TCGA ovarian cancer patients mRNA expression datasets. These findings further support our conclusions that ERβ agonist mediated cell cycle arrest and apoptosis may involve upregulation of FDXR. However, further in-depth mechanistic studies are needed to address the specific roles of FDXR in ERβ mediated function in OCSCs.

## 4. Materials and Methods

### 4.1. Cell Lines and Reagents

Human ovarian cancer (OCa) cell lines, SKOV3 (Cat # HTB-77), OV90 (Cat # CRL-11732), and ES2 (Cat # CRL-1978), were obtained from the American Type Culture Collection (ATCC, Manassas, VA, USA) and maintained as per ATCC guidelines. The OVSAHO (Cat # SCC294) cell line was obtained from Sigma Chemical Co, St. Louis, MO, USA. A2780 cell line was a generous gift from Dr. Komaraiah Palle (Texas Tech University Health Sciences Center, Lubbock, TX, USA). Cell identity was confirmed using short tandem repeat polymorphism (STR) DNA profiling. Cell lines were maintained in a humidified chamber with 5% CO_2_ at 37 °C. Following standard laboratory protocols, all model cells utilized were determined to be free of mycoplasma contamination by using Mycoplasma PCR Detection Kit (Sigma, St. Louis, MO, USA). p21 Waf1/Cip1 (Cat # 2947, 1:1000), CD133 (Cat # 64326, 1:1000), SOX2 (Cat # 3579, 1:1000), Oct4 (Cat # 2890, 1:1000), Nanog (Cat # 4903, 1:1000), PARP (Cat # 9542, 1:1000), cleaved caspase 3 (Cat # 9661, 1:1000), and GAPDH (Cat # 8884, 1:1000) antibodies were purchased from Cell Signaling Technology (Beverly, MA, USA). FDXR (Cat # HPA044393, 1:1000) antibody was purchased from Sigma (St. Louis, MO, USA). Anti-mouse (Cell Signaling, Cat # 70765, 1:1000) and anti-rabbit (Cytiva Life Sciences, Marlborough, MA, USA, Cat# NA934, 1:2000) secondary antibodies were purchased from manufacturers, respectively. ERβ antibody (Cat # GTX70174, ICC 1:100) was purchased from GeneTex (Irvine, CA, USA). ERβ agonist LY500307 was purchased from Cayman Chemical (Ann Arbor, MI, USA). 

### 4.2. Isolation of OCSCs and Cell Viability Assays

Ovarian cancer stem cells (OCSCs) were enriched from ES2, OV90, SKOV3, OVSAHO, and A2780 cells with ALDEFLUOR kit (STEMCELL Technologies, Cambridge, MA, USA; Cat # 01700) using BD FACSAria cell sorter. OCSCs were cultured as spheroids in DMEM-F12 medium supplemented with B27 serum free supplement, EGF (20 ng/mL), bFGF (20 ng/mL), hydrocortisone (0.5 μg/mL), and insulin (5 μg/mL). DEAB that inhibits the ALDH activity was used as a negative control to set the gate of the ALDH positive (ALDH+) population. Representative flow cytometry images defining ALDH-ve and ALDH+ve cells were shown in Appendix A. For ERβ isoforms expression analysis, ALDH+ve and ALDH-ve cells were collected immediately after cell sorting and subjected to RNA isolation. For cell viability assays, ALDH-ve cells were cultured in DMEM-F12 supplemented with 10% FBS. The effect of ERβ agonist LY500307 on the viability of OCSCs was determined using Cell Titer Glo luminescence assay as described earlier [33]. Briefly, ALDH+ and ALDH—cells were seeded in 96 well plates and after overnight incubation, cells were treated with varying doses of LY500307. After 96 h, Cell-Titer-Glo reagent was added to each well, and luminescence was measured using GloMax Luminometer (Promega, Madison, WI, USA).

### 4.3. Spheroid Formation and Extreme Limiting Dilution Assays

The effect of ERβ agonist LY500307 on sphere formation was determined using spheroid assays. Briefly, single cell suspensions of OCSCs were seeded in 24-well plates (100 cells/well) in triplicates and treated with vehicle or LY500307 (3 µM). After 7 days, the number of newly formed spheres was counted, and the size of the sphere was determined using NIH Image J software. The effect of LY500307 on self-renewal ability of OCSCs was determined using extreme limiting dilution assays (ELDA). Briefly, OCSCs were plated in decreasing numbers (50, 20, 10, 5, and 1 cell(s)/well) in 96 well ultra-low attachment plates and treated with vehicle or LY500307. After 10 days, the number of wells containing spheres per plating density was counted. Stem cell frequency between vehicle and LY500307 treated groups was calculated using ELDA analysis software (https://bioinf.wehi.edu.au/software/elda/, accessed on 25 October 2021).

### 4.4. Annexin V, Cell Cycle, and Cell Invasion Assays

The effect of LY500307 on apoptosis of OCSCs was determined using Annexin V/PI assay kit (BioLegend, San Diego, CA, USA; Cat # 640914) [69]. Briefly, OCSCs were treated with either vehicle (0.1% DMSO) or LY500307 (5 µM) for 48 h. Then, 100 µL of the cell suspension was incubated with Annexin V FITC and propidium iodide (PI) for 15 min at room temperature in the dark. Annexin V binding buffer (400 µL) was then added to each sample and stained cells were analyzed using flow cytometry. For cell cycle assay, OCSCs were treated with either vehicle (0.1% DMSO) or LY500307 (5 μM) for 72 h. Cells were harvested and subjected to fixation in ice-cold 70% ethanol for 30 min at 4 °C. Cells were then washed with PBS and stained with a mixture of 100 μg/mL propidium iodide and 50 μg/mL RNase A. The cell cycle distribution of OCSCs was then determined by using flow cytometry. The effect of LY500307 on the invasion of OCSCs was determined using Corning^®^ BioCoat™ Growth Factor Reduced Matrigel Invasion Chamber assay (Corning, NY, USA; Cat # 354480) as per the manufacturer’s instructions.

### 4.5. Cell Lysis and Western Blotting 

Whole cell lysates from SKOV3-ALDH+ and A2780-ALDH+ OCSCs were prepared using RIPA buffer (Thermo Fisher Scientific, Waltham, MA, USA, Cat # 89901) containing protease and phosphatase inhibitors. Total lysates were mixed with 4X SDS sample buffer, and run on SDS-Polyacrylamide gels. Proteins were transferred to PVDF membrane and subsequently incubated in 5% non-fat dry milk powder in TBST (Bio-Rad, Hercules, CA, Cat # 1706404) for 1 h at room temperature. Primary antibody incubation at indicated dilutions, occurred overnight at 4 °C with gentle agitation. Blots were subjected to secondary antibody incubation (anti-mouse 1:1000 and anti-rabbit 1:2000) for 1 h at room temperature and developed using the ECL kit (Millipore, Burlington, MA) using radiographic film or BioRad Chemidoc MP. GAPDH antibody was used as loading control. Quantitation was performed using NIH ImageJ software.

### 4.6. RNA-Sequencing

SKOV3-ALDH+ OCSCs were treated with vehicle or LY500307 (5 µM) for 24 h and the total RNA was isolated using RNeasy Mini Kit (Qiagen, Valencia, CA; Cat# 74104). Illumina TruSeq stranded mRNA-seq library preparation and sequencing were performed by UTHSA Genome Sequencing Facility as described previously [70]. DEseq2 was used to identify differentially expressed genes (DEGs) between the groups. DEGs with a criteria of *p*-value < 0.05 and abs(log2Foldchange) > 0.5 were chosen for further analysis. Gene set enrichment analysis (GSEA) was performed on all genes identified by RNA-seq to study genes enrichment in a specific function. The sequenced data were deposited in the GEO database under a GEO accession number GSE197320. 

### 4.7. RT-qPCR

For RT-qPCR assays, cDNA was synthesized from RNA using a High-capacity cDNA reverse transcription kit (Applied Biosystems, Foster City, CA, USA). RT-qPCR was performed using a PowerUp SYBR Green Master Mix (Applied Biosystems) on a StepOnePlus Real-Time PCR System using gene-specific primer sequences obtained from Harvard Primer Bank (https://pga.mgh.harvard.edu/primerbank/, accessed on 27 September 2021). Data were normalized to GAPDH, and the difference in fold change was calculated using delta-delta-CT method. Primers sequences used for ERβ isoforms 1, 2, 4, and 5 are as follows: ERβ1-forward: 5′-GTCAGGCATGCGAGTAACAA-3′; ERβ1-reverse: 5′-GGGAGCCCTCTTTGCTTTTA-3′; ERβ2-forward: 5′-TCTCCTCCCAGCAGCAATCC-3′; ERβ2-reverse: 5′-GGTCACTGCTCCATCGTTGC-3′; ERβ4-forward: 5′-GTGACCGATGCTTTGGTTTG-3′; ERβ4-reverse: 5′-ATCTTTCATTGCCCACATGC-3′; ERβ5-forward: 5′-GATGCTTTGGTTTGGGTGAT-3′; ERβ5-reverse: 5′-CCTCCGTGGAGCACATAATC-3′. Primer sequences for other genes used in the study were provided in Appendix A.

### 4.8. Reporter Gene Assays

ERE reporter gene assays were performed as described earlier [59]. SKOV3 cells were stably transfected with an estrogen response element (ERE) reporter plasmid (System Biosciences, Palo Alto, CA, USA) and cultured as spheroids. Cells were then treated with vehicle or LY500307 for 24 h and lysed in luciferase lysis buffer. The luciferase activity was measured by using the Luciferase Assay system with a GloMax Luminometer (Promega, Madison, WI, USA). 

### 4.9. Immunocytochemistry (ICC)

ICC experiments were performed as described earlier [59]. Briefly, ALDH-ve and ALDH+ve cells enriched from SKOV3 and A2780 cells were seeded on fibronectin coated slides in 24 well plates and after 24 h cells were fixed with 4% paraformaldehyde. Cells were permeabilized with 0.5% Triton X-100 followed by washing with phosphate-buffered saline. Cells were then subjected to blocking with 5% bovine serum albumin followed by incubation with ERβ primary antibody (1:100) overnight. Cells were subsequently incubated with HRP-conjugated secondary antibody for 1 h at room temperature. Immunoreactivity was detected by using the diaminobenzidine (DAB) substrate and counterstained with hematoxylin (Vector Lab, Inc, Burlingame, CA, USA.). Slides were imaged using a Nikon Ti inverted microscope.

### 4.10. In Silico Exploration of ESR2 Target Genes

Motifs of ERβ/ESR2 were acquired from JASPAR (Matrix ID: MA0258.1; MAO258.2) [71], and mapped with the FIMO and T-Gene modules from the MEME Suite in sequence [72] to identify potential binding targets of ERβ. Then those potential target genes were intersected with LY500307-regulated DEGs from our RNAs-seq results. Correlation analysis between target genes and ERβ were performed with GEPIA2 using TCGA OV tumor dataset [73]. ERβ ChIP-seq profile from CistromeDB (CistromeDB ID: 34675) was used for preliminary validation of ERβ binding in the promoters of potential target genes. 

### 4.11. Mice Studies

Animal experiments were conducted according to institutional guidelines and prior IACUC approval. NCr/SCID female mice of 8 weeks of age were purchased from Charles River Laboratories (Wilmington, MO). SKOV3-ALDH+ cells were labeled with GFP-Luc and treated with vehicle or LY500037 (5 μM) for 72 h and the viable cells at serial dilutions (1 × 10^3^, 1 × 10^4^, 1 × 10^5^, and 1 × 10^6^) were injected into the ovarian bursa (*n* = 4/group). Tumor initiation potential of vehicle and LY500037 treated cells were monitored for 30 days using Xenogen In vivo Imaging System. In vivo tumorigenic frequency between vehicle and LY500307 treated groups was calculated using ELDA analysis software (https://bioinf.wehi.edu.au/software/elda/, accessed on 20 December 2021).

### 4.12. Statistical Analyses

Statistical differences were analyzed with unpaired Student’s *t*-test and one-way ANOVA using GraphPad Prism 8 software. All data presented in plots are shown as mean ± SE. A value of *p* < 0.05 was considered statistically significant. 

## 5. Conclusions

In summary, our study demonstrated that ERβ1 is highly expressed in OCSCs. ERβ agonist treatment, potentially via upregulation of FDXR and CDKN1A, promotes apoptosis and reduces stemness of OCSCs. We surmise that ERβ agonist LY500307 may represent a promising therapeutic for the treatment of OCa.

## Figures and Tables

**Figure 1 ijms-23-07159-f001:**
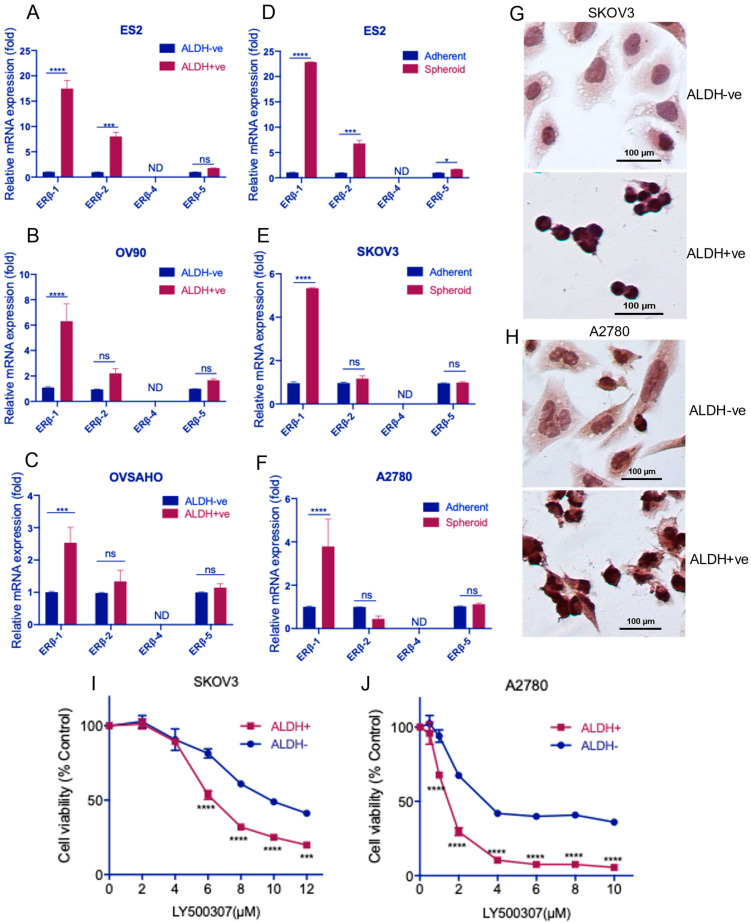
ERβ is highly expressed in OCSCs. (**A**–**C**)**,** ES2, OV90, and OVSAHO cells were subjected to ALDEFLUOR assay and the expression of ERβ isoforms 1, 2, 4, and 5 in ALDH-ve and ALDH+ve cells was determined using RT-qPCR. (**D**–**F**), SKOV3, ES2, and A2780 cells were cultured as adherent (2D) and spheroid (3D) cultures and the expression of ERβ isoforms 1, 2, 4, and 5 was determined using RT-qPCR. (**G**,**H**), ALDH-ve and ALDH+ve cells enriched from SKOV3 and A2780 cells were seeded on fibronectin coated plates and after 24 h cells were fixed in 4% paraformaldehyde and subjected to ICC staining with ERβ antibody as described in methods. (**I**,**J**), ALDH-ve and ALDH+ve populations isolated from SKOV3 and A2780 cells were treated with vehicle or indicated concentrations of LY500307 for 96 h, and the cell viability rates were determined using Cell-Titer-Glo assay. Data are represented as mean ± SE. ns: no significance; * *p* < 0.05; *** *p* < 0.001; **** *p* < 0.0001.

**Figure 2 ijms-23-07159-f002:**
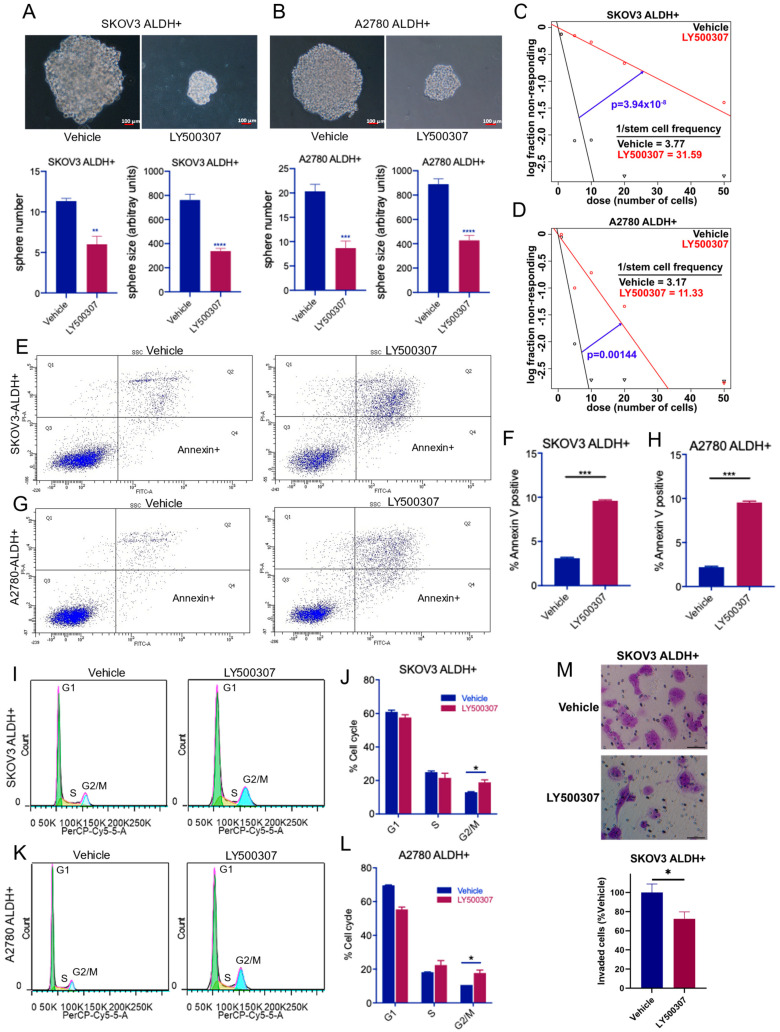
ERβ agonist LY500307 reduced sphere formation and self-renewal, and promoted the apoptosis and cell cycle arrest of OCSCs. (**A**,**B**) SKOV3- and A2780- ALDH+ve cells were seeded triplicates in 24 well plates and treated with vehicle or LY500307 for 7 days. Representative images of the resulting spheres in each treatment are shown. The number of spheres was counted manually. Sphere size was determined using ImageJ software. Accompanying quantitation is shown below. (**C**,**D**) SKOV3- and A2780- ALDH+ve cells were seeded in decreasing concentrations and treated with vehicle or LY500307. After 10 days, the sphere formation in each plating density was determined and self-renewal rates were analyzed using ELDA software. (**E**–**H**) SKOV3 and A2780 OCSCs were treated with LY500307 for 48 h and its effect on apoptosis was determined using Annexin V/PI assay. (**I**–**L**) SKOV3- and A2780- OCSCs were treated with LY500307 for 72 h and the cell cycle distribution was determined using flow cytometry. (**M**) SKOV3 OCSCs were treated with vehicle or LY500307 for 48 h, and their invasion ability was determined using matrigel invasion chamber assay. Data are represented as mean ± SE. * *p* < 0.05; ** *p* < 0.01; *** *p* < 0.001; **** *p* < 0.0001.

**Figure 3 ijms-23-07159-f003:**
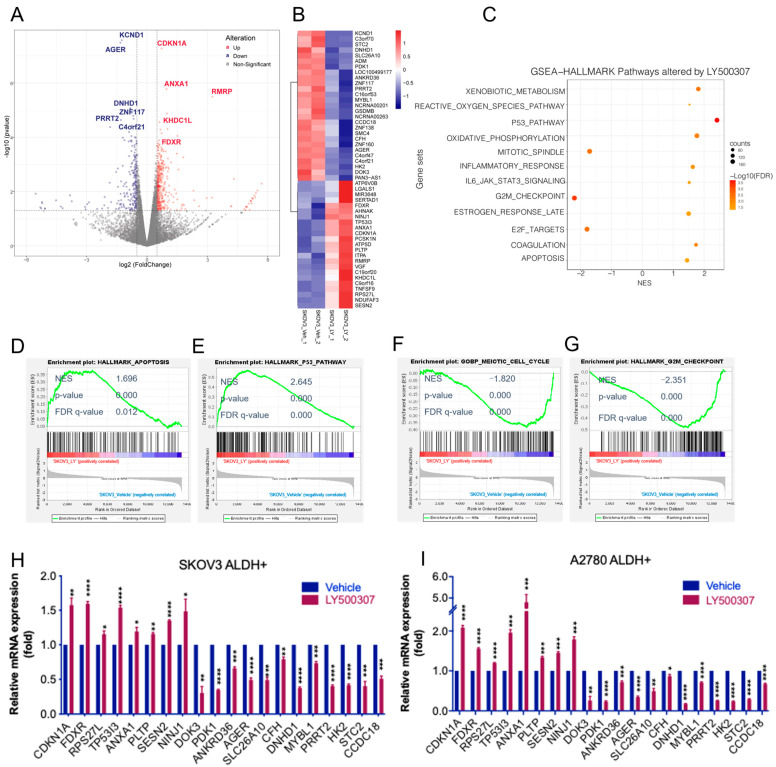
Identification of altered pathways following ERβ agonist treatment using RNA seq analysis. SKOV3 ALDH+ve cells were treated with vehicle or LY500307 for 24 h, RNA was isolated and then subjected to RNA sequencing. (**A**) Volcano plots comparing gene expression levels for vehicle vs. LY500307 treated OCSCs. Counts values were normalized with DESeq2, X-axis showed the Log2 fold change of gene expression levels between groups, and Y-axis indicates -log10(*p*-value). Significantly upregulated or downregulated genes were marked in red and blue, respectively, with the criteria of *p* < 0.05 and |log2(fold change)| > 0.5. The top 10 DEGs, including CDKN1A and FDXR, are labeled. (**B**) Heatmap showing top up- and downregulated DEGs. (**C**) Bubble plot of top GSEA hallmark gene set enrichment results. Y-axis showed the gene set names; X-axis indicated the normalized enrichment scores (NES). Positive NES indicates gene set enrichment in LY500307 treated group, while negative NES indicates gene set enrichment in Vehicle treated group. Dot size indicates the gene number enriched. Color demonstrates –log10(FDR), showing the significance of enrichment. (**D**–**G**) GSEA of LY500307 modulated genes shows correlation with signatures of apoptosis, p53 pathway, cell cycle, and G2/M checkpoint gene sets. (**H**,**I**) SKOV3- and A2780- ALDH+ve cells were treated with vehicle or LY500307 for 24 h, and selected genes involved in p53 pathway, apoptosis, and cell cycle progression were validated using RT-qPCR. Data are represented as mean ± SE. * *p* < 0.05; ** *p* < 0.01; *** *p* < 0.001; **** *p* < 0.0001. NES: normalized enrichment score; FDR: false discovery rate.

**Figure 4 ijms-23-07159-f004:**
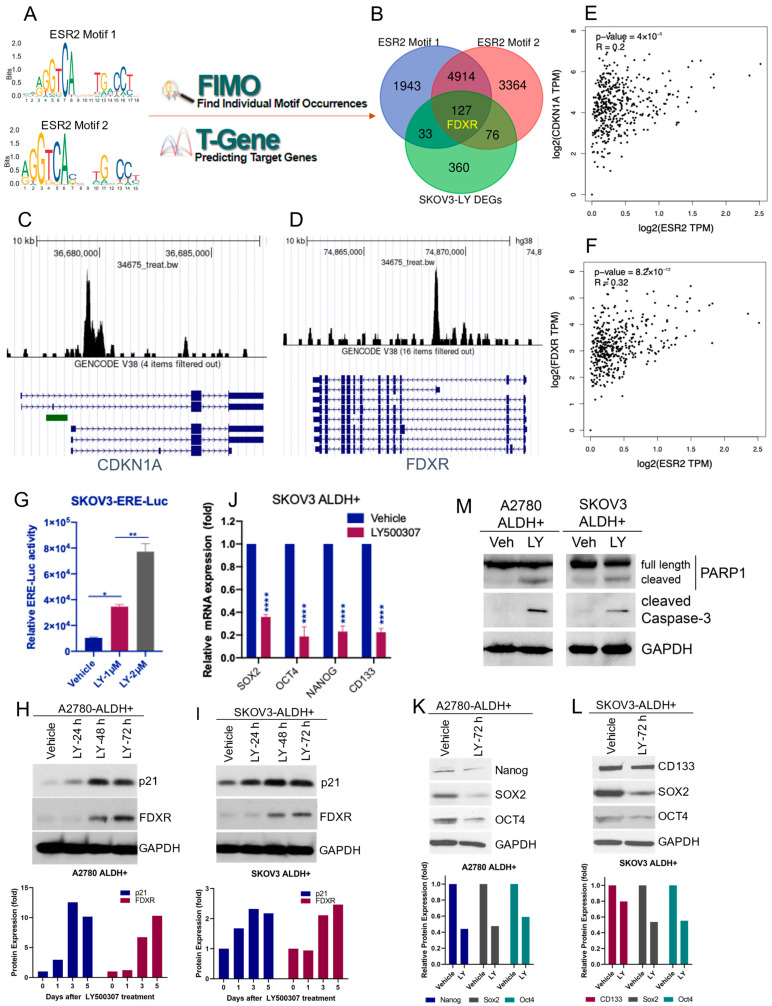
In silico analysis suggested FDXR and CDKN1A could be direct targets of ERβ. (**A**) Exploration of potential ERβ target genes. Motifs of ESR2 (ERβ) were acquired from JASPAR and mapped with the FIMO and T-Gene modules from the MEME Suite in sequence. (**B**) Potential ERβ target genes were identified by the intersection of potential target genes with LY500307-regulated DEGs in OCSCs, including FDXR. (**C**,**D**) ChIP-seq profiles from CistromeDB confirmed enrichment of ERβ at CDKN1A and FDXR gene promoters. (**E**,**F**) Co-expression analysis using GEPIA2 showed CDKN1A and FDXR expression are positively correlated with ESR2 expression. (**G**) SKOV3 cells stably expressing ERE-luc reporter plasmid were cultured as spheroids for 7 days and treated with vehicle or LY500307 for 24 h. ERE-luc reporter activity was measured using a Luciferase Assay System. (**H**,**I**) SKOV3- and A2780- ALDH+ve cells were treated with LY500307 and harvested at 24 h, 48 h, and 72 h after treatment and p21 and FDXR expression was determined by Western blotting. Relative protein band intensity of representative blots was quantitated using NIH ImageJ software and shown in bottom panels. (**J**) SKOV3 ALDH+ve cells were treated with vehicle or LY500307 for 72 h and stemness marker expression was determined using RT-qPCR. (**K**–**M**) SKOV3- and A2780- ALDH+ve cells were treated with vehicle or LY500307, and expression of stemness (Nanog, SOX2, OCT4, and CD133) and apoptotic markers (cleaved PARP1 and Caspase-3), respectively, was determined using Western blotting. Data are represented as mean ± SE. * *p* < 0.05; ** *p* < 0.01; **** *p* < 0.0001.

**Figure 5 ijms-23-07159-f005:**
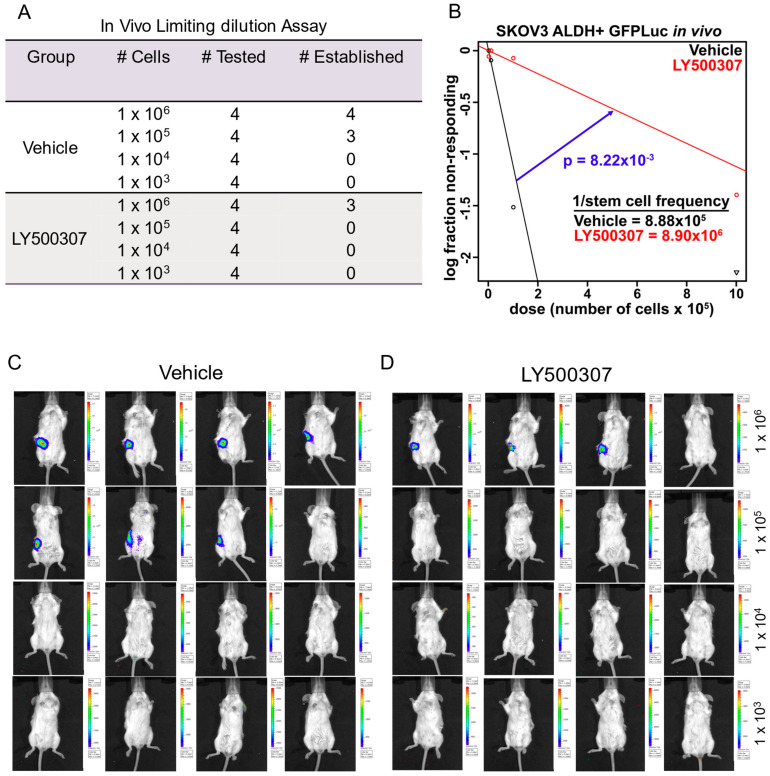
LY500307 treatment reduced the tumor initiation capacity of OCSCs. SKOV3 ALDH+ve cells that stable expressing GFP-Luc were treated with vehicle or LY500307 for 72 h, and the varying number of viable cells were injected into ovarian bursa. Mice were monitored for tumor establishment for 30 days and imaged weekly using Xenogen in vivo imaging system. (**A**) Table showing the mice per each group and number of mice with tumor establishment. (**B**) ELDA analysis was performed to estimate the tumorigenic capacity of OCSCs. (**C**,**D**) Pictures of the mice at the end of the experiment were shown.

## Data Availability

The RNA-seq data were deposited in the GEO database under a GEO accession number GSE197320.

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
