# Peer review of "Therapeutic Targeting of Ovarian Cancer Stem Cells Using Estrogen Receptor Beta Agonist"

_ijms, 2022, doi:10.3390/ijms23137159_

Round 1

Reviewer 1 Report

In this study, authors have demonstrated that ERβ1 is highly expressed in OCSCs. ERβ agonist treatment, in part via upregulation of FDXR and CDKN1A, promotes apoptosis and reduces stemness of OCSCs. They also showed that ERβ agonist LY500307 may represent a promising therapeutic for the treatment of ovarian cancer. 

There are a few minor comments that needed to be addressed.

1.    Fig-5C, luminescence color code intensity scale bar is missing.

2.    Correlation co-efficient value 0.2 is very poor to appreciate in Fig-2E. Please employ one or two other publicly available dataset to ensure the association between ESR2 versus CDKN1A.

Author Response

Reviewer 1:

Comment:  Fig-5C, luminescence color code intensity scale bar is missing.

Response:  We thank the reviewer for this suggestion. We have now added color code intensity scale bar for luminescence images.

Comment:   Correlation co-efficient value 0.2 is very poor to appreciate in Fig-2E. Please employ one or two other publicly available dataset to ensure the association between ESR2 versus CDKN1A.

Response:  We thank the reviewer for this comment. We believe that the low coefficient value is due to complexity of ER beta isoforms present in Ovarian tumors, lack of specific probes to correlate the data exclusively with the full length ERbeta1 isoform, and the heterogeneity of ovarian tumor samples used in the public data. We did try a couple of other databases to examine the correlation of ESR2 and CDKN1A and found that correlation co-efficient is still below 0.3. However, this correlation is statistically significant with p-val <0.0001.

Reviewer 2 Report

In this manuscript, the authors demonstrated that: 1) ALDH+ ovarian cancer stem cells (CSCs) express higher levels of ER-beta1 expression than ALDH- non-CSC, and 2) treatment of ALDH+ ovarian CSC with ER-beta1 agonist LY500307 decreases self-renewal and tumor-initiating capacity of ALDH + CSC cells in vitro and in vivo. The authors also identified FDXR and p21 as direct targets of ER-beta1. Overall, this manuscript is well written, and data are clearly presented. However, there are several areas that need to be improved to strengthen the authors’ findings. Please see below for details. Several images are pixelated (e.g. Fig 2, Fig 3), and fonts are too small to read (e.g. Fig 3). Please make sure to use high quality images and increase font sizes according to the journal’s guidelines. Also please provide the supplementary tables showing 596 DEG genes (line 168) and 127 candidate genes (line 207).

ER-beta is highly expressed in normal ovary and its expression is significantly reduced during ovarian cancer progression (line 94). Then when and how does ovarian CSC exhibit the expression of ER-beta? Do ER-beta1-expressing ovarian CSCs appear at a later stage or after chemotherapy treatment? Or does chemotherapy induce ER-beta expression in ovarian CSC? The authors also mentioned that the decrease in ER-beta expression correlates with a lack of clinical response to chemotherapy and shorter overall survival in OC (line 69-70). Since ovarian CSC is also associated with a lack of clinical response to chemotherapy (or chemoresistance) and shorter overall survival, this clinical observation seems to contradict the authors’ observation that ovarian CSCs express higher ER-beta than bulk cancer cells. Please discuss these points in the discussion section. It would be also helpful to add references if there are any studies to show that chemotherapy-resistant ovarian cancer patients express higher levels of ER-beta1 than chemo-sensitive patients to support the authors’ findings.

Since normal ovary also expresses ER-beta, would this cause some toxicity/side effects when ovarian cancer patients are treated with LY500307? Did the authors observe any abnormalities or cell death in ovaries and oviducts of LY500307-treated mice?

Line 147 Fig 2I-L: Please provide the percentage of cells in G2/M phase between vehicle and LY500307-treated CSC as well as p values to see if this difference is statistically significant.

Line 125. Please indicate what protein was stained in Fig 1G-H in the figure legend.

Fig 2A-B: please include scale bars. Fig 2E-G: Please change the label to SKOV3 ALDH+ and A2780 ALDH+ since the current labeling causes the misunderstanding that the authors treated bulk cancer cells with LY500307.

Line 371: Please indicate what machine was used to isolate/sort ALDH + and ALDH- cells. Please indicate the company and catalog number of ALDEFLUOR kit. Please provide the gating of ALDH positive and negative A2780 and ES2 cells with or without DEAB in the supplementary figure to show how the authors defined ALDH positive and negative cells in A2780 and ES2 cells.

Line 399: Please indicate the company and catalog number of Annexin V kit.

Line 417: Please provide what dilution was used for primary and secondary antibodies in WB. What is the loading control?

Line 430. Please provide a list of primer sequences used in Fig 3H-I as well as GAPDH and beta actin in the supplementary table.

Line 445: Please indicate the source of the ERE reporter construct.

Line 455: Please provide what dilution was used for primary and secondary antibodies in ICC.

Line 483: Since the authors did not provide any functional data to show that FDXR and CDKN1A are responsible for ER-beta1 mediated ovarian CSC phenotype, it might be better to replace ‘in part’ with ‘potentially’.

Line 486: There were no supplementary materials associated with this manuscript. Please revise the section accordingly.

Line 500: Since the authors deposited RNA-Seq data and obtained the accession number, please revise this section accordingly.

Author Response

Reviewer 2:

Comment: Several images are pixelated (e.g. Fig 2, Fig 3), and fonts are too small to read (e.g. Fig 3). Please make sure to use high quality images and increase font sizes according to the journal’s guidelines.

Response: We have provided the high quality images for figure 2 and figure 3 and increased the font sizes whereever appropriate.

Comment: Also please provide the supplementary tables showing 596 DEG genes (line 168) and 127 candidate genes (line 207).

Response: We have provided the supplementary tables depicting the 596 DEGs (Supplementary table 1) as well as 127 potential ERbeta target genes (Supplementary table 2).

Comment: ER-beta is highly expressed in normal ovary and its expression is significantly reduced during ovarian cancer progression (line 94). Then when and how does ovarian CSC exhibit the expression of ER-beta? Do ER-beta1-expressing ovarian CSCs appear at a later stage or after chemotherapy treatment? Or does chemotherapy induce ER-beta expression in ovarian CSC? The authors also mentioned that the decrease in ER-beta expression correlates with a lack of clinical response to chemotherapy and shorter overall survival in OC (line 69-70). Since ovarian CSC is also associated with a lack of clinical response to chemotherapy (or chemoresistance) and shorter overall survival, this clinical observation seems to contradict the authors’ observation that ovarian CSCs express higher ER-beta than bulk cancer cells. Please discuss these points in the discussion section. It would be also helpful to add references if there are any studies to show that chemotherapy-resistant ovarian cancer patients express higher levels of ER-beta1 than chemo-sensitive patients to support the authors’ findings.

Response: We thank the reviewer for this insightful comment. The exact mechanism that regulates ERbeta1 expression in ovarian CSCs remains unknown. We speculate that the contradictory ERbeta expression in bulk tumor cells vs. cancer stem cells could be due to altered expression of ERbeta isoforms and tissue-specific function of ERbeta. Further, some evidence implicated the lack of clinical response to chemotherapy due to decreased ERbeta1 expression, which is attributed in part due to increased expression of other isoforms due to altered splicing of full-length ERbeta1 to ERbeta2, ERbeta5 that contribute to therapy resistance. To the best of our knowledge, there are no studies that examined the ERbeta expression in chemotherapy-resistant ovarian cancer vs. chemo-sensitive patients. However, a  recent study showed that A high level of ERβ expression in stage III HGSOC patients can predict the efficacy of platinum plus taxane chemotherapy. We discussed these points in the discussion section.

Comment: Since normal ovary also expresses ER-beta, would this cause some toxicity/side effects when ovarian cancer patients are treated with LY500307? Did the authors observe any abnormalities or cell death in ovaries and oviducts of LY500307-treated mice?

Response: We thank the reviewer for this comment. Our earlier studies showed that ERbeta agonist showed less toxicity to normal astrocytes which express ERbeta when compared to glioblastoma cell lines( PMID: 27126081). Interestingly, ER beta agonist LY500307 was also well tolerated in BPH patients with no side effects (PMID: 25348255) and has also been tested in phase 2 clinical trials for improving negative symptoms and cognitive impairment associated with Schizophrenia (http://clinicaltrials.gov/show/NCT01874756). Our unpublished data also showed that ERbeta agonist has significantly lower cytotoxicity to normal fallopian tube epithelial cells when compared to ovarian cancer cell lines. We believe that ERbeta actions are cell specific with ERbeta is being protective in normal cell cells while inducing cell death in cancer cells.

Comment: Line 147 Fig 2I-L: Please provide the percentage of cells in G2/M phase between vehicle and LY500307-treated CSC as well as p values to see if this difference is statistically significant.

Response: We have now provided the percentage of cells in G2/M phase folllowing vehicle or LY500307 treatment and LY500307 treatment significantly arrested the cells in G2/M phase when compared to vehicle treatment.

Comment: Line 125. Please indicate what protein was stained in Fig 1G-H in the figure legend.

Response: We have provided the antibody information in the figure legends.

Comment: Fig 2A-B: please include scale bars. Fig 2E-G: Please change the label to SKOV3 ALDH+ and A2780 ALDH+ since the current labeling causes the misunderstanding that the authors treated bulk cancer cells with LY500307.

Response: We have provided the scalebars for Fig2A-B and also changed the labels to SKOV3-ALDH+ and A2780-ALDH+ in Fig. 2E-G.

Comment: Line 371: Please indicate what machine was used to isolate/sort ALDH + and ALDH- cells. Please indicate the company and catalog number of ALDEFLUOR kit. Please provide the gating of ALDH positive and negative A2780 and ES2 cells with or without DEAB in the supplementary figure to show how the authors defined ALDH positive and negative cells in A2780 and ES2 cells.

Response: We have provided the FACS sorter information, catalogue information of ALDEFLOUR kit. We have also provided the gating images of ALDH+ and ALDH- cells with and without DEAB in the supplementary figure.

Comment: Line 399: Please indicate the company and catalog number of Annexin V kit.

Response: Company and catalog number of Annexin V kit are now included.

Comment: Line 417: Please provide what dilution was used for primary and secondary antibodies in WB. What is the loading control?

Response: We have provided the information on primary and secondary antibody dilutions under reagents section as well as loading control antibody.

Comment: Line 430. Please provide a list of primer sequences used in Fig 3H-I as well as GAPDH and beta actin in the supplementary table.

Response: We have provided the primer sequences in the supplementary table 3.

Comment: Line 445: Please indicate the source of the ERE reporter construct.

Response: We included ERE-luc reporter construct information.

Comment: Line 455: Please provide what dilution was used for primary and secondary antibodies in ICC.

Response: We have provided the antibody dilutions for ICC experiment.

Comment: Line 483: Since the authors did not provide any functional data to show that FDXR and CDKN1A are responsible for ER-beta1 mediated ovarian CSC phenotype, it might be better to replace ‘in part’ with ‘potentially’.

Response: We have modified the sentence as suggested.

Comment: Line 486: There were no supplementary materials associated with this manuscript. Please revise the section accordingly.

Response: We have revised the manuscript as suggested.

Comment: Line 500: Since the authors deposited RNA-Seq data and obtained the accession number, please revise this section accordingly.

Response: We have revised these sections accordingly.

Reviewer 3 Report

In this study, Authors have analyzed the status of estrogen receptor beta ERβ expression in Ovarian Cancer Stem Cells (OCSCs) and the therapeutic utility of the ERβ agonist LY500307 for targeting OCSCs. They have enriched OCSCs from five OC cells and ascertained by RT-qPCR that ERβ isoform 1, is highly expressed in OCSCs compared to non-OCSCs and that ERβ agonist LY500307 significantly reduced the viability of OCSCs.

The study appears well conducted using proper and convincing methods, and the results of treatment of OCSCs with LY500307 support the rationale, demonstrating that this ERβ agonist is highly effective in reducing the stemness and promoting apoptosis of OCSCs.

They conclude that LY500307shows significant promise as a novel therapeutic agent in treating OCa.

My only criticism is precisely with regard to how LY500307 was presented, who despite being the protagonist of this research, is not in the least introduced, presented neither in the introduction nor in the whole manuscript, without even a reference. Is there a brief history of this compound? Is it possible to have a brief summary in the Introduction, with the relative reference(s)?

Author Response

Comment:

My only criticism is precisely with regard to how LY500307 was presented, who despite being the protagonist of this research, is not in the least introduced, presented neither in the introduction nor in the whole manuscript, without even a reference. Is there a brief history of this compound? Is it possible to have a brief summary in the Introduction, with the relative reference(s)?

Response: We thank the reviewer for this suggestion. We have now provided a brief summary of the LY500307 compound in the introduction section.

LY500307 (SERBA1, erteberel) is a potent (EC50 = 0.66 nM) and selective synthetic ERβ agonist developed by Eli Lilly and Company (PMID: 17034120). LY500307 has a 14-fold higher affinity for ERβ than ERα and exhibits 32-fold more in functional assays. LY500307 was well tolerated with no side effects (PMID: 25348255) and has also been tested in phase 2 clinical trials for improving negative symptoms and cognitive impairment associated with Schizophrenia (http://clinicaltrials.gov/show/NCT01874756). Further, LY500307 was shown to exhibit antitumor functions in glioblastoma models (PMID: 27126081, PMID: 33470499) and activation of ERβ with LY500307 augments innate immunity to suppress lung metastasis of triple negative breast cancer and melanoma cells (PMID: 29592953). However, the role of ERβ in OCSCs and the effect of ERβ agonists on OCSC functions remain largely unknown.

Round 2

Reviewer 2 Report

In the revised manuscript, the authors have addressed the reviewer's comments to improve the clarity of the manuscript. The authors have also provided all necessary supplementary data. The reviewer does not have any further suggestions and recommend this manuscript for publication.